# Alkaline Water Splitting by Ni-Fe Nanoparticles Deposited on Carbon Fibre and Nickel-Coated Carbon Fibre Substrates

Mateusz Kuczyński [ID], Tomasz Mikołajczyk *[ID] and Bogusław Pierożyński *

Department of Chemistry, Faculty of Agriculture and Forestry, University of Warmia and Mazury in Olsztyn, Łódzki Square 4, 10-727 Olsztyn, Poland; mateusz.kuczynski@uwm.edu.pl
* Correspondence: tomasz.mikolajczyk@uwm.edu.pl (T.M.); boguslaw.pierozynski@uwm.edu.pl or bogpierozynski@yahoo.ca (B.P.)

**Abstract:** This study presents the results of electrochemical investigations on Hydrogen and Oxygen Evolution Reactions (HER and OER), conducted on commercially available carbon fibres and nickel-coated carbon fibres modified using nanoscale NiFe alloy particles in 0.1 M of NaOH solution. The obtained results demonstrated enhanced catalytic activity of the NiFe-modified fibre materials, with approximately 14,700% and 25% improvement in the OER and HER activity (respectively), as compared to unmodified electrodes. The catalytic properties were evaluated by means of electrochemical impedance spectroscopy, Tafel polarisation and cyclic, and linear voltammetry techniques. The deposited particles' distribution and quantities present on the investigated materials were analysed using Scanning Electron Microscopy (SEM) and Energy-Dispersive X-ray spectroscopy (EDX) methods. These findings provided valuable insights into the electrochemical, catalytic performance of NiFe-modified carbon fibre/nickel-coated carbon fibre materials, simultaneously highlighting their potential application as catalyst materials for electrodes in industrial-scale water electrolysers.

**Keywords:** HER; OER; Ac. impedance spectroscopy; transition metals; electrocatalysts; water electrolysis; electrodeposition; NiFe-nanomaterials

## 1. Introduction

As the world faces an energy crisis and serious challenges associated with climate change, researchers continue searching for new solutions and alternatives in order to replace conventional fossil fuels [1–3]. Hydrogen's high energy density makes it a promising alternative fuel solution. Moreover, its combustion results in the formation of just water molecules, which is accompanied by energy release in the form of heat (Equation (1)). Unfortunately, a significant portion (95%) of hydrogen is currently produced by means of methods that rely on fossil fuels (methane, coal, and oil), through steam methane reforming or coal gasification, leading to significant carbon dioxide emissions. This type of hydrogen is commonly known as "gray hydrogen". On the other hand, water electrolysis (e.g., realised via PEM: proton exchange membrane or AWE: alkaline water electrolysis process) combined with renewable energy sources (e.g., solar, wind or water) enables hydrogen production to become emission-free, where such generated $H_2$ is called "green hydrogen" [4–6].

$$2H_2 + O_2 \rightarrow 2H_2O + Q \tag{1}$$

Due to the acidic environment prevailing inside the PEM electrolyser cell, these systems could only utilise catalysts that are highly stable (corrosion-resistant) at low pH values, i.e., predominantly composed of very costly, semi-noble, and noble metal particles. In contrast, the alkaline (AWE) systems enable the utilisation of cheaper and more available transition elements, such as Fe, Ni, Co, Cu, etc. However, these metals are generally less catalytic towards either of the discussed gas evolution reactions than those of noble/semi-noble nature. This necessitated a search for new materials that could lower hydrogen

production costs by reducing the overall energy consumption for the water electrolysis process [7–10]. Current AWE scientific activities are primarily focused on the investigation of the catalytic performance of different forms of Ni, Fe, and Co elements. Of particular interest are materials derived from their respective oxides and hydroxides, as well as various alloy compositions containing these elements. Thus, their chemical compositions and synthesis methods are being continuously analysed and are subject to ongoing discussions within the scientific community [11–20].

Recently, special attention has been given to NiFe-LDH (NiFe Layered Double Hydroxide)-based materials, which exhibit superior catalytic properties and achieve high current densities towards the HER and OER reactions. Additionally, notable materials, such as NiSn, NiCoSn, NiCu, MoS, NiAg, and CoP demonstrate similar catalytic properties to Pt in the context of the HER [11,20–24]. Furthermore, when analysing the OER catalysts, it is worthwhile (besides the NiFe alloy) to consider materials such as CoP, NiCoO, and NiCo-LDH, which also demonstrate promising catalytic properties [14,17,18].

This work aims to shed light on the behaviour of NiFe catalysts on commercially available carbon fibre (CF) and nickel-coated carbon fibre (NiCCF) materials, and serves as a preliminary investigation for further studies, enabling the practical implementation of NiFe alloys in industrial-scale electrolysers. The authors' decision to combine Ni-coated carbon fibres (or carbon fibres) with NiFe alloy was inspired by the potential of leveraging the inherent electrochemical properties of both materials. While NiCCF and CF offer robust and conductive support materials, NiFe alloy particles further enhance their catalytic (HER, OER) activities. It should also be stressed here that the current work takes direct advantage of a number of previously published articles from this laboratory on the HER behaviour of CF and NiCCF tow catalyst materials (see Refs. [25–29]). Specifically, the CF (NiCCF) and NiFe catalyst combination aims to address the challenges associated with single-material systems and seeks to optimize the performance of both the HER and OER reactions under alkaline conditions.

## 2. Results and Discussion

### 2.1. SEM/EDX Characterisation of CF, NiCCF, NiFe/CF and NiFe/NiCCF Electrodes

Figures 1a–c and 2a–c show that the structures of the deposited NiFe alloy were of somewhat non-regular and discontinuous structure based on the base material's surface. The grain size of the deposited alloy fluctuated between 40–60 nm with visible agglomerates of NiFe particles reaching sizes up to 150 nm. NiFe alloy content in the NiFe/CF samples was ca. 10 wt.% (assessed by a weighing method and the SEM coupled with EDX spectroscopy evaluations). The weighing method showed the same results for the NiFe/NiCCF sample, as for the NiFe/CF specimen. For the SEM/EDX analysis, notable discrepancies were observed in the percentage composition of individual elements (see Table 1). The above could be attributed to the presence of a homogeneous nickel coating, which impairs the accessibility of carbon during the SEM/EDX examination. The total composition and arrangement of elements of the NiFe/CF, NiCCF, and NiFe/NiCCF electrodes is shown in Table 1 and Figures 1 and 2.

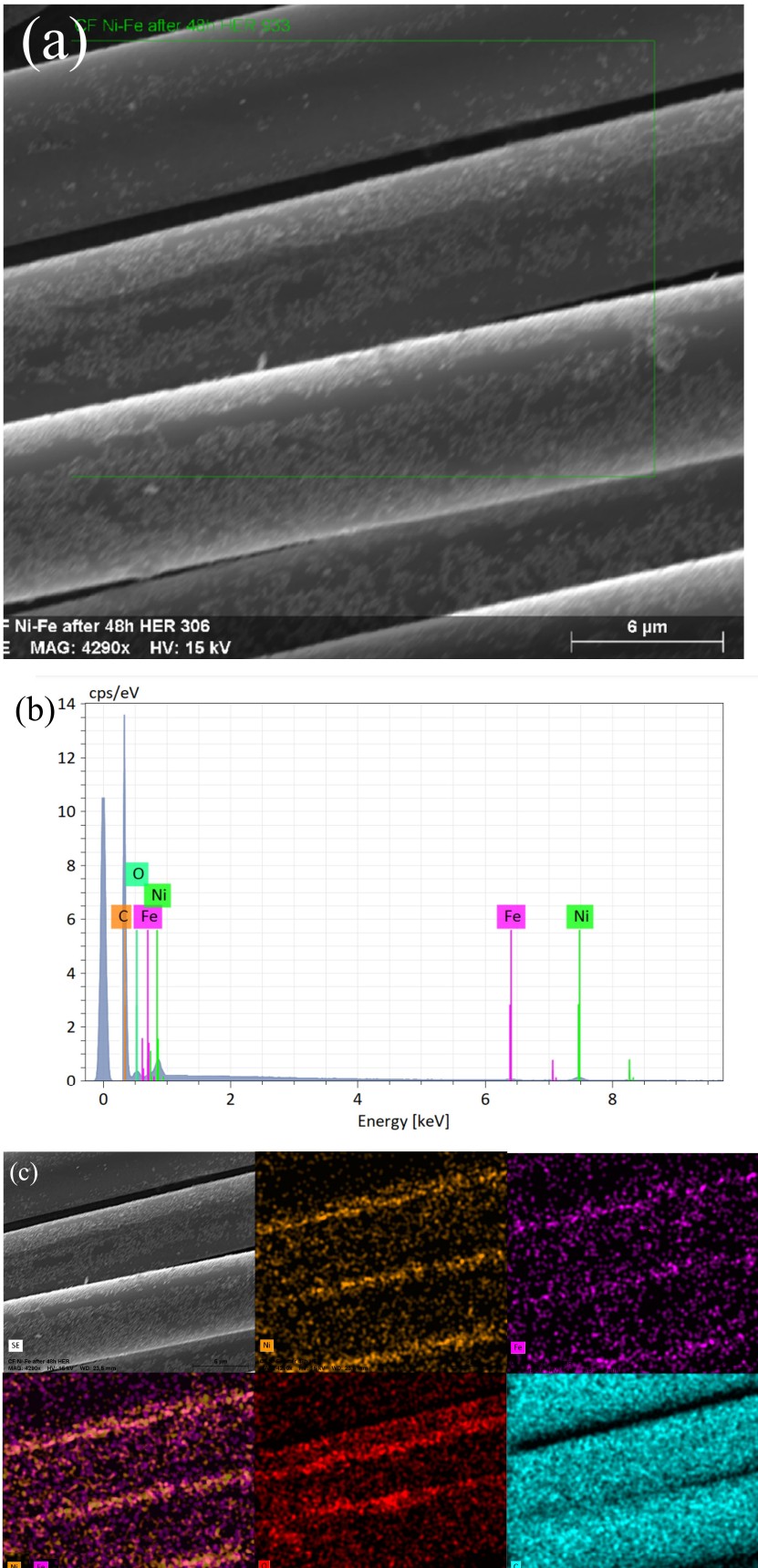

**Figure 1.** SEM micrograph pictures of NiFe/CF (**a**), taken at 4290× magnification with EDX pattern (**b**) and EDX elemental mappings (**c**).

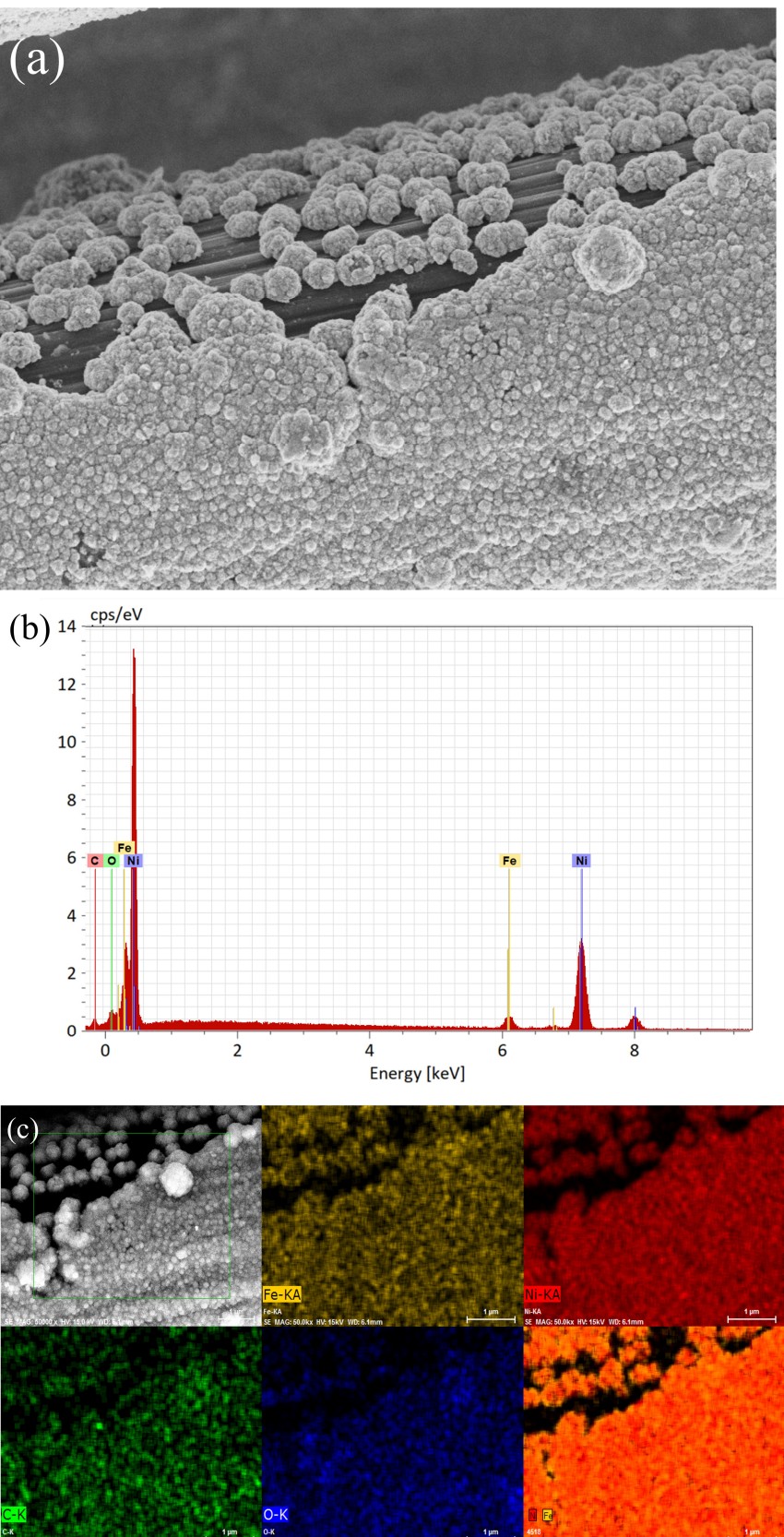

**Figure 2.** SEM micrograph pictures of NiFe/NiCCF (**a**), taken at 50,000× magnification with EDX pattern (**b**) and EDX elemental mappings (**c**).

**Table 1.** EDX-derived (for an acceleration voltage of 15 kV) chemical composition of surface elements for NiFe/CF, NiCCF, and NiFe/NiCCF samples.

| | NiFe/CF | | |
|---|---|---|---|
| Element | Spectrum 1 [wt.%] | Spectrum 2 [wt.%] | Spectrum 3 [wt.%] |
| C | 76.33 | 76.91 | 74.65 |
| O | 13.19 | 14.07 | 12.88 |
| Ni | 8.78 | 7.79 | 10.50 |
| Fe | 1.69 | 1.24 | 1.97 |
| | NiCCF | | |
| Element | Spectrum 1 [wt.%] | Spectrum 2 [wt.%] | Spectrum 3 [wt.%] |
| C | 5.07 | 4.72 | 5.28 |
| O | 0.77 | 1.01 | 1.30 |
| Ni | 94.16 | 94.27 | 93.42 |
| | NiFe/NiCCF | | |
| Element | Spectrum 1 [wt.%] | Spectrum 2 [wt.%] | Spectrum 3 [wt.%] |
| C | 2.85 | 3.73 | 2.66 |
| O | 1.28 | 1.46 | 2.53 |
| Ni | 91.28 | 88.96 | 90.21 |
| Fe | 4.59 | 5.85 | 4.60 |

*2.2. Electrochemical Characterisation*

2.2.1. Cyclic Voltammetry

The cyclic voltammetry (CV) graphs present a comparison of electrochemical behaviour for all examined electrodes (CF, NiCCF, NiFe/CF, and NiFe/NiCCF) in 0.1 M NaOH solution (three sweeps were carried out over the potential span of $-1.0$–1.8 V vs. RHE with a scan-rate of 50 mV s$^{-1}$—the last cycles are presented) in Figure 3a,b. The deposition of NiFe alloy on the surfaces of CF and NiCCF materials resulted in a significant enhancement of the HER and OER catalysis. Additionally, the recorded cyclic voltammograms for the NiFe/CF electrodes exhibited two anodic (A, B) and three cathodic (C, D, E) peaks (see marked peaks in Figure 3: A (0–700 mV), B (1400–1700 mV), C (1000–1500 mV), D (350–700 mV) and E ($-100$–300 mV)). Peak A corresponds to the oxidation of iron (Equations (2)–(4)) and nickel (Equations (5)–(7), where $\alpha$-Ni(OH)$_2$ ageing is applied; see Bode cycle diagram in Figure 12 of Ref. [30] for more details) along with the corresponding reduction peaks D (Fe$^{3+}$/Fe$^{2+}$) and E [Fe$^{2+}$/Fe$^0$ and Ni(OH)$_2$/Ni$^0$]. On the other hand, peak B is related to the formation of $\beta$-NiOOH oxyhydroxide phase (Equation (8)); peak C corresponds to its reduction ($\beta$-NiOOH/$\beta$-Ni(OH)$_2$) [31–38]. However, as no ageing was applied in this work to in situ formed nickel hydroxide, its significant portion would further be converted upon charging to form $\gamma$-NiOOH phase (Equation (9)). Hence, the recorded peaks B/C in Figure 3a most likely correspond to mixed features of $\beta$-NiOOH/$\beta$-Ni(OH)$_2$ and $\gamma$-NiOOH/$\alpha$-Ni(OH)$_2$ transitions.

$$Fe^0 \rightarrow Fe^{2+} + 2e^- \tag{2}$$

$$Fe^0 + 2OH^- \rightarrow Fe(OH)_2 + 2e^- \tag{3}$$

$$Fe^{2+} + 3OH^- \rightarrow Fe(OH)_3 + e^- \tag{4}$$

$$Ni^0 + 2OH^- \rightarrow \alpha\text{-}Ni(OH)_2 + 2e^- \tag{5}$$

$$\alpha\text{-Ni(OH)}_2 \rightarrow \beta\text{-Ni(OH)}_2 \tag{6}$$

$$Ni^0 + 2OH^- \rightarrow \beta\text{-Ni(OH)}_2 + 2e^- \tag{7}$$

$$\beta\text{-Ni(OH)}_2 + OH^- \rightarrow \beta\text{-NiOOH} + H_2O + e^- \tag{8}$$

$$\alpha\text{-Ni(OH)}_2 + OH^- \rightarrow \gamma\text{-NiOOH} + H_2O + e^- \tag{9}$$

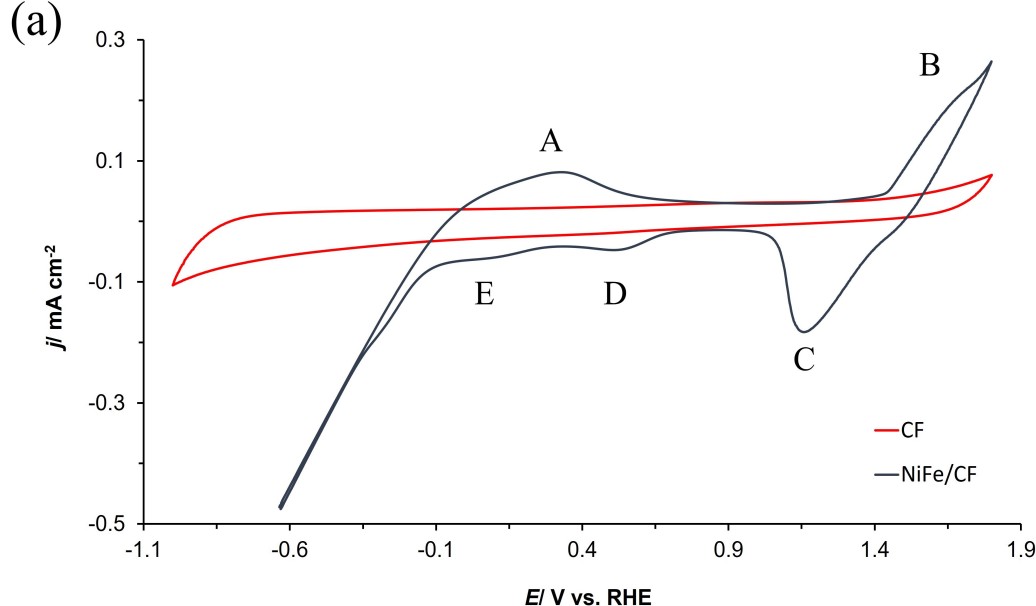

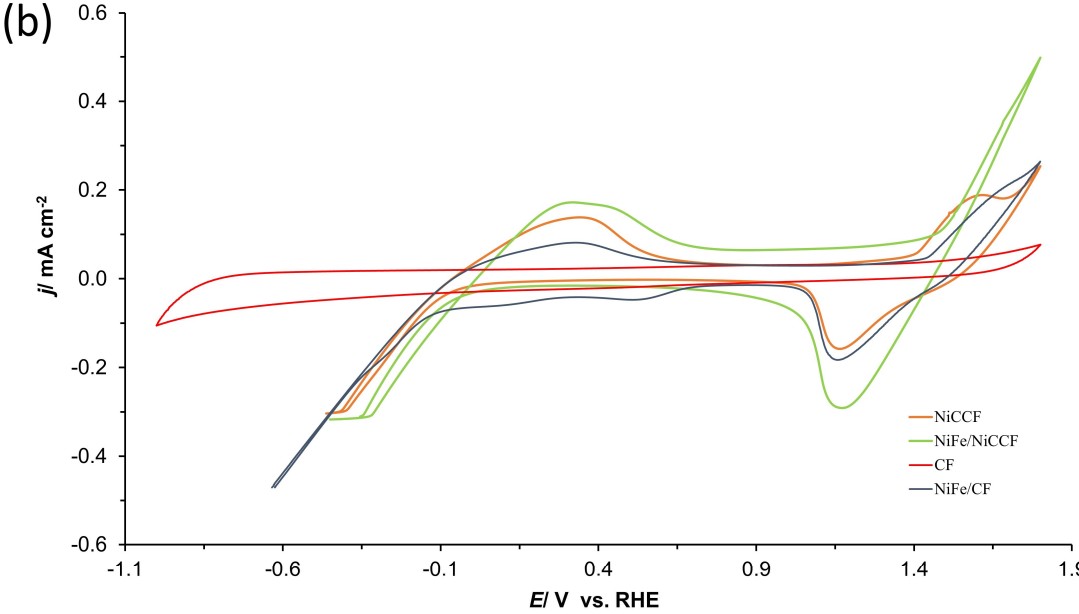

**Figure 3.** Cyclic voltammogram curves of (**a**) CF and NiFe/CF; (**b**) CF, NiCCF, NiFe/CF, and NiFe/NiCCF electrodes in contact with 0.1 M NaOH medium, carried out at a scan-rate of 50 mV s$^{-1}$ over the potential span from −1.0 to 1.8 V vs. RHE.

Figure 4 presents the cyclic voltammetry (CV) curves for all examined fibre-based electrodes. The comparison reveals noticeable differences among the samples. Specifically, as expected for the unmodified NiCCF electrode, no cathodic peaks (peaks D and E) corresponding to iron reduction are observed there. Additionally, it could be noticed that the current densities recorded on the NiFe/NiCCF electrode for peaks A and C are significantly higher than those obtained on the NiFe/CF sample. However, in the case of the NiFe/NiCCF sample, peak B is hardly visible. Most importantly, the presence of NiFe alloy significantly reduces overpotentials for the OER and HER processes. Furthermore, the NiFe (at 10 wt.%)/CF electrode exhibits similar OER/HER catalytic properties to those demonstrated by commercially manufactured NiCCF products, with an average Ni content at 45 wt.% (see Figure 3b and Table 2 below for more details and corresponding data in SF: Figure S2 and Table S1).

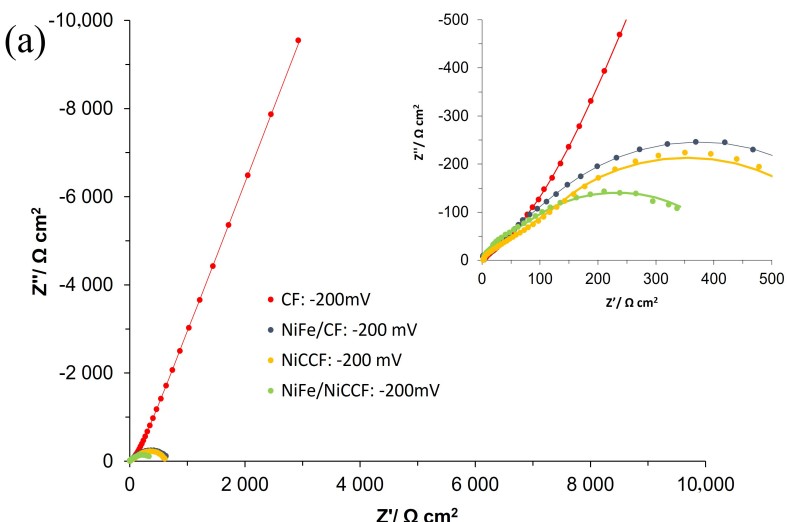

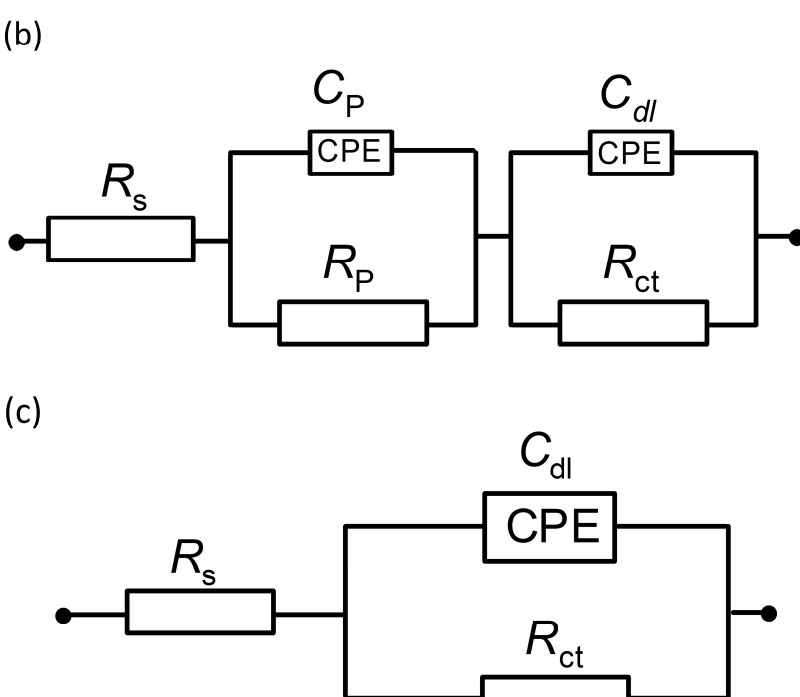

**Figure 4.** *Cont.*

(d)

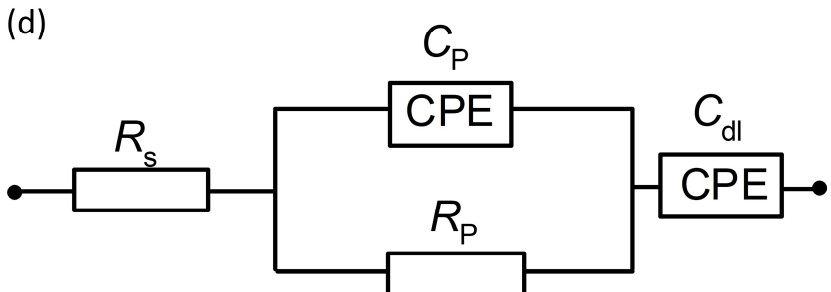

**Figure 4.** (a) Electrochemical impedance Nyquist plots for the HER on CF, NiCCF, NiFe/CF, and NiFe/NiCCF electrode surfaces in contact with 0.1 M NaOH (at 293 K) for the potential of −200 mV vs. RHE; (**b**–**d**) equivalent circuits used to fit the above process, where $C_p$ is the Faradaic pseudo-capacitance, $R_p$ is the Faradaic resistance and $C_{dl}$ is the double-layer capacitance (both capacitance parameters are CPE: constant phase element−modified), jointly in series with an uncompensated solution resistance, $R_s$. The data derived from the equivalent circuits are represented by the solid lines.

**Table 2.** Current densities for HER ($\eta = 0.35$ V) and OER ($\eta = 0.57$ V) recorded from CV curves.

| Sample | Current Densities [mA cm$^{-2}$] | |
|:---:|:---:|:---:|
| | **HER$_{\eta=0.35V}$** | **OER$_{\eta=0.57V}$** |
| CF | −0.042 | 0.077 |
| NiFe/CF | −0.222 | 0.264 |
| NiCCF | −0.253 | 0.254 |
| NiFe/NiCCF | −0.309 | 0.498 |

### 2.2.2. A.c Impedance-HER

Figure 4a (and corresponding Figure S3) and Table 3 show the impedance spectroscopy results for modified electrodes and electrodes made of base materials, examined in 0.1 M NaOH. The electrochemical parameters, such as charge transfer resistance ($R_{ct}$), porosity resistance for reaction intermediates ($R_p$), double-layer capacitance ($C_{dl}$), and pseudo-capacitance ($C_p$) parameters were obtained using two constant phase element (CPE)—modified Randles equivalent circuit model (Figure 4b–d). The impedance measurements of unmodified electrodes (CF) for potentials (from −100 to −600 mV vs. RHE) showed one depressed semicircle (associated with porosity response at high frequency) and linear part of the plot corresponding to CPE-modified capacitive response, recorded at medium and low frequencies. Then, between the potentials of −700 and −900 mV vs. RHE, a second semicircle corresponding to HER becomes visible on the EIS plot (medium and low frequencies). The $R_p$ and $C_p$ parameters presented for pure CF are mostly potentially independent as they could be associated with a response similar to the porous surface, simulated by the tow material [39]. In contrast, the $C_{dl}$ parameter increased correspondingly from 83.8 to 214.3 μF cm$^{-2}$ for the potentials of −100 and −900 mV. This phenomenon is probably associated with very poor catalytic properties and a strongly electrochemicaly non-uniform surface of the CF electrode; thus, increasing surface area becomes activated along with rising overpotential [39,40]. The $R_{ct}$ parameter (observed in the range of −700 to −900 mV) decreased from 19,156.8 to 1780.7 Ω cm$^2$, respectively, which is characteristic of the kinetically controlled potential ranges.

**Table 3.** Electrochemical parameters for the HER, obtained at as received CF, NiFe/CF, NiCCF, and NiFe/NiCCF electrodes in contact with 0.1 M NaOH supporting solution. The results obtained here were recorded by fitting the CPE-modified Randles equivalent circuit (the superscripts attached to potential value correspond to the model from Figure 4b–d) to the experimentally obtained impedance data (reproducibility usually below 10%, $\chi^2 = 1.56 \times 10^{-6}$ to $1.74 \times 10^{-5}$).

| $E/\text{mV}$ | $R_\text{p}/\Omega \text{ cm}^2$ | $C_\text{p}/\mu\text{F cm}^{-2}$ | $R_\text{ct}/\Omega \text{ cm}^2$ | $C_\text{dl}/\mu\text{F cm}^{-2}$ |
|---|---|---|---|---|
| **CF** | | | | |
| −100 [d] | 91.7 ± 0.1 | 435.3 ± 28.0 | - | 83.8 ± 0.1 |
| −200 [d] | 107. 1 ± 0.1 | 642.2 ± 48.4 | - | 88.2 ± 0.2 |
| −300 [d] | 127.6 ± 0.1 | 954.8 ± 79.4 | - | 94.4 ± 0.2 |
| −400 [d] | 144.2 ± 0.1 | 914.4 ± 64.2 | - | 104.3 ± 0.3 |
| −500 [d] | 149.6 ± 0. 1 | 867.3 ± 56.5 | - | 116.0 ± 0.4 |
| −600 [d] | 112.7 ± 3.8 | 495.5 ± 61.7 | - | 134.2 ± 0.3 |
| −700 [b] | 86.2 ± 4.5 | 652.2 ± 7.9 | 19,156.8 ± 715.9 | 154.9 ± 1.0 |
| −800 [b] | 71.5 ± 3.9 | 620.2 ± 10.1 | 4403.1 ± 56.9 | 176.1 ± 1.4 |
| −900 [b] | 66.7 ± 5.8 | 352.8 ± 7.8 | 1780.7 ± 26.9 | 214.3 ± 3.2 |
| **NiFe/CF** | | | | |
| −100 [b] | 222.5 ± 11.4 | 834.8 ± 12.0 | 1572.9 ± 15.5 | 325.3 ± 2.1 |
| −200 [b] | 333.9 ± 89.4 | 996.4 ± 86.1 | 405.6 ± 61.8 | 363.4 ± 22.3 |
| −300 [c] | - | - | 370.9 ± 6.1 | 287.4 ± 11.7 |
| −400 [c] | - | - | 263.5 ± 10.6 | 363.6 ± 29.8 |
| −500 [c] | - | - | 189.0 ± 5.2 | 378.1 ± 28.2 |
| −600 [c] | - | - | 142.2 ± 4.9 | 338.9 ± 34.7 |
| −700 [c] | - | - | 120.7 ± 6.8 | 371.5 ± 57.2 |
| **NiCCF** | | | | |
| −100 [b] | 169.1 ± 5.8 | 113.2 ± 3.3 | 745.4 ± 10.3 | 137.5 ± 2.0 |
| −200 [b] | 132.6 ± 6.0 | 108.4 ± 4.1 | 490.3 ± 8.6 | 119.6 ± 2.4 |
| −300 [b] | 149.7 ± 11.1 | 91.3 ± 3.7 | 244.9 ± 12.7 | 118.2 ± 5.4 |
| −400 [b] | 130.1 ± 11.2 | 161.7 ± 39.9 | 213.2 ± 11.2 | 129.8 ± 15.1 |
| −500 [c] | - | - | 202.9 ± 6.0 | 111.2 ± 4.1 |
| −600 [c] | - | - | 214.8 ± 6.7 | 110.9 ± 4.2 |
| **NiFe/NiCCF** | | | | |
| −100 [b] | 68.2 ± 4.8 | 172.5 ± 16.3 | 563.7 ± 12.3 | 446.7 ± 9.6 |
| −200 [c] | - | - | 454.2 ± 10.4 | 363.7 ± 16.7 |
| −300 [c] | - | - | 306.5 ± 9.8 | 321.4 ± 23.7 |
| −400 [c] | - | - | 217.6 ± 5.8 | 221.6 ± 17.8 |
| −500 [c] | - | - | 215.0 ± 31.2 | 259.9 ± 64.7 |

The impedance Nyquist plots for the NiFe/CF electrodes showed two depressed semicircles in the potential range of −100 to −200 mV, where the high-frequency semicircle corresponds to the porosity of the electrode, and the low-frequency semicircle is related to the kinetics of the hydrogen evolution reaction. Notably, the presence of a semicircle connected to the HER process at lower overpotentials for modified electrodes suggests that these electrodes possess higher catalytic activity compared to the base CF electrode. The semicircle corresponding to $R_\text{p}$ and $C_\text{p}$ parameters was no longer visible for more

negative potentials, as the CF tow material spread due to extended formation of $H_2$ bubbles, thus losing its somewhat porous structure. The value of the $R_{ct}$ parameter was radically reduced—by about 160 times, as compared to the $R_{ct}$ values obtained for unmodified CF at the potential of $-700$ mV. Also, the NiFe modification caused the value of $C_{dl}$ parameter to increase by 2.4 times for the same potential value. These results show that the presence of NiFe alloy significantly improves the CF material's catalytic properties towards the HER. It is important to note that when focusing solely on the catalytic effect, independent of surface area changes, the enhancement of electrochemical performance is primarily driven by the catalytic properties of the NiFe alloy surface modifier (*ca.* 67 times, excluding the surface area augmentation).

Also, for the NiFe/CF electrode, increasing cathode overpotentials steadily caused the $R_{ct}$ parameter to be reduced from 1572.9 $\Omega$ cm$^2$ to 120.7 $\Omega$ cm$^2$ for the tested potential range. However, no significant changes were observed in the $C_{dl}$ parameter values with rising cathode overpotentials. The fluctuation in the $C_{dl}$ parameter values could be associated with the simultaneous blocking of the electrode surface by freshly formed $H_2$ bubbles and "opening" of the CF tow material by these bubbles, thus leading to increased accessibility to the electrode's surface area [29].

The EIS measurements for unmodified NiCCF resulted in two distinct semicircles in the potential range of $-100$ to $-400$ mV. The values of $R_p$ and $C_p$ parameters were independent of the applied potential and ranged between 130.1, 168.1 $\Omega$ cm$^2$ and 91.3, and 161.7 $\mu$F cm$^{-2}$, respectively. However, similarly to the EIS response for the NiFe/CF electrode, the $C_p$ and $R_p$ parameters were no longer visible at higher overpotentials. Similarly to the NiFe/CF catalyst material, the $R_{ct}$ parameter for the NiCCF electrode showed a decreasing trend with increasing overpotential, while the $C_{dl}$ showed some unspecific fluctuations. Compared to the NiCCF electrode, the NiFe-modified CF catalyst exhibited considerably lower $R_{ct}$ parameter values, but primarily at significant cathodic overpotentials (Table 3).

The modification of the NiCCF electrode with NiFe alloy reduced the charge transfer resistance parameter by approximately 25% at the electrode potential of $-100$ mV. In contrast, the values of the $C_{dl}$ parameter for the modified electrodes increased by approximately three times, as compared to unmodified ones. This suggests that the modification primarily influences the active surface area of the electrodes rather than its catalytic properties. Also, the behaviour of the $R_{ct}$ and the $C_{dl}$ parameters for the NiFe-modified nickel-coated carbon fibre electrodes with rising overpotentials was similar to that observed for the unmodified ones; however, the recorded $C_{dl}$ values for the former case were somewhat reduced, as compared to those derived for the latter ones.

The relationship of $-\log R_{ct}$ and overpotential ($\eta$) for kinetically controlled reactions was selected here over the potential range $-100$ to $-600$ mV vs. RHE with 100 mV potential increments. The exchange current densities, $j_0$, were calculated for the HER based on the Butler–Volmer equation and the relation between the $j_0$ and the $R_{ct}$ parameter for the overpotential approaching 0 (see Equation (10) below) [26].

$$j_0 = \frac{RT}{zFR_{ct}} \tag{10}$$

Such calculated $j_0$ reached the values of $1.8 \times 10^{-11}$, $1.7 \times 10^{-6}$, $1.4 \times 10^{-6}$, and $1.7 \times 10^{-6}$ A cm$^{-2}$ for CF, NiFe/CF, NiCCF, and NiFe/NiCCF catalyst samples, respectively. The $j_0$ values for the NiFe/CF, NiCCF, and NiFe/NiCCF electrodes showed comparable results, where the observed differences in the $j_0$ values between the NiFe/CF and NiFe/NiCCF samples were indeed insignificant. This indicates that even with a lower catalyst (NiFe) loading, the samples achieved similar activities to that exhibited by the commercial product, which contains over four times as much Ni catalyst. The results presented in Table 4 (and corresponding Table S2) suggest that NiFe alloy deposited on carbon fibre is more cost-effective, compared to the same base material, but activated by noble metals. This conclusion can be drawn based on the comparable performance of NiFe to Pd- or Ru-modified carbon fibres (see Table 4 for details). Therefore, the utilisation of NiFe alloy

provides a more economically viable alternative for diverse energy-related applications, including the oxidation of organic compounds, such as urea to serve as a more efficient and cost-effective anode option for the production of electrolytic hydrogen energy carrier, as compared to traditional water electrolysis strategies. Additionally, NiFe alloy could also serve as a superior active material for electrochemical supercapacitors. Its remarkable electrical conductivity and high surface area jointly contribute to efficient energy storage and charge/discharge characteristics, improving the supercapacitors' overall performance and durability [41–43].

**Table 4.** Exchange current densities for the HER (calculated based on the Butler–Volmer equation) in 0.1 M NaOH.

| Material | HER | |
| :---: | :---: | :---: |
| | $j_0$ [A cm$^{-2}$] | Ref. |
| CF | $1.8 \times 10^{-11}$ | This work |
| NiFe/CF | $1.7 \times 10^{-6}$ | This work |
| NiCCF | $1.4 \times 10^{-6}$ | This work |
| NiFe/NiCCF | $1.7 \times 10^{-6}$ | This work |
| Ru/NiCCF | $5.4 \times 10^{-5}$ | [44] |
| Pd/CF | $1.7 \times 10^{-5}$ | [28] |
| Ru/CF | $7.7 \times 10^{-6}$ | [40] |

### 2.2.3. Tafel-HER

The Tafel polarisation plots recorded for CF, NiFe/CF, NiCCF, and NiFe/NiCCF electrodes are shown in Figure 5 (corresponding to Figure S4). The recorded cathodic slopes ($b_c$) and exchange current densities for the HER are presented in Table 5. The potential range in which these parameters were measured was −50 to −200 mV for the NiFe/CF, NiCCF, and NiFe/NiCCF samples. However, as the onset of hydrogen evolution was observed at much more negative potentials on the CF sample, the potential range −700 to −900 mV/RHE was chosen for this electrode. This phenomenon was also reflected in the EIS results. The values of the Tafel-plot-derived $j_0$ parameter are similar to those obtained by means of the Butler–Volmer equation-based method and demonstrate significantly improved catalytic properties after the NiFe electrode modification. Furthermore, these electrodes exhibited a more positive onset potential, as compared to the unmodified ones in Figure 6 (and related Figure S5). Furthermore, the obtained values of the exchange current density are comparable with the literature values presented in Tables 4 and 5 (and corresponding Table S3).

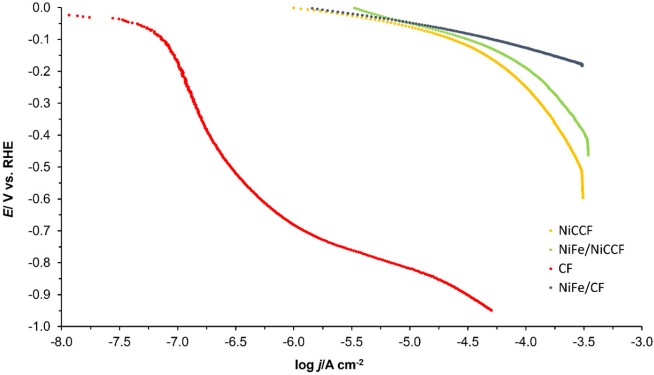

**Figure 5.** Quasi-potentiostatic cathodic polarisation curves for the hydrogen evolution reaction (HER), obtained at CF, NiFe/CF, NiCCF, and NiFe/NiCCF electrodes in 0.1 M NaOH electrolyte. The polarisation curves were recorded at a scanning rate of 0.5 mVs$^{-1}$. The impedance-based solution resistance, i$R$, correction was also applied.

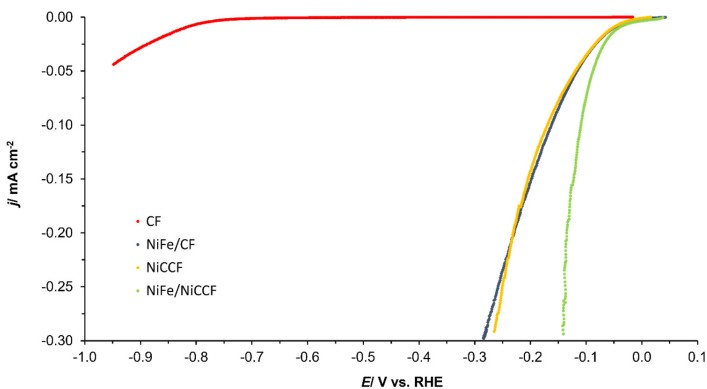

**Figure 6.** Linear Sweep Voltammetry (LSV) curves of CF, NiFe/CF, NiCCF, and NiFe/NiCCF electrodes in 0.1 M NaOH solution, carried out with a scan rate of 0.5 mV s$^{-1}$ for the HER (*iR*-corrected).

Then, for carbon-based electrodes, modified with 10 wt.% of NiFe alloy, the recorded $j_0$ for the HER approached those typically derived for unmodified nickel electrodes [45]. Nevertheless, their catalytic efficiency towards the HER is about six times lower than that recorded for platinum electrodes. Interestingly, it is possible to find transition metal alloys that closely approach their HER parameters and the performance of platinum or even exhibit superior behaviour to the Pt (including one that is based on a NiFe catalyst [21,46]). This implies that the NiFe catalysts evaluated in this work may need to be optimised in order to enhance their HER performance.

**Table 5.** HER kinetic parameters for the selected catalytic materials.

| Material | $b_c$ [mV dec$^{-1}$] | $j_0$ [A cm$^{-2}$] | Ref. | Electrolyte |
|:---:|:---:|:---:|:---:|:---:|
| CF | $-108$ | $3.1 \times 10^{-13}$ | This work | 0.1 M NaOH |
| NiFe/CF | $-62$ | $1.7 \times 10^{-6}$ | This work | 0.1 M NaOH |
| NiCCF | $-63$ | $1.5 \times 10^{-6}$ | This work | 0.1 M NaOH |
| NiFe/NiCCF | $-67$ | $1.2 \times 10^{-6}$ | This work | 0.1 M NaOH |
| NiFe/NiFoam | 157 | $1.7 \times 10^{-5}$ | [46] | 1.0 M KOH |
| Ni | - | $2.3 \times 10^{-6}$ | [45] | 0.1 M NaOH |
| Pt | $-150$ | $1.0 \times 10^{-5}$ | [47] | 0.1 M NaOH |
| NiSn/Cu | $-121$ | $6.9 \times 10^{-7}$ | [11] | 1.0 M KOH |
| NiCoSn/Cu | $-122$ | $1.2 \times 10^{-5}$ | [11] | 1.0 M KOH |
| NiCu/C | $-57$ | $2.5 \times 10^{-5}$ | [21] | 1.0 M KOH |

### 2.2.4. A.c. Impedance-OER

The impedance spectroscopy results for all examined electrode types are shown in Table 6. The OER behaviour presented in Figure 7 demonstrates that introducing modifications to the based carbon fibre electrodes resulted in considerably increasing the reactivity of the tested electrodes. The $R_p$ and $C_p$ parameters independently fluctuate in the span of the applied electrode potentials (also, see an explanation of the behaviour of these parameters in Section 2.2.2). The catalytic modification in the case of the CF electrode caused the recorded $R_p$ value to decrease from 150.7 to 41.6 $\Omega$ cm$^2$ (at the potential of 1400 mV), while the $C_p$ value increased from 276.0 to 1982.0 μF cm$^{-2}$ at the same electrode potential. On the other hand, the $R_{ct}$ and $C_{dl}$ parameters' values strongly depended on the applied potential. Specifically, the $R_{ct}$ parameter decreased along with increasing potential. In comparison, a decrease in the $C_{dl}$ parameter upon the potential augmentation was

slightly less pronounced (probably caused by a stronger blocking effect of $O_2$ bubbles, compared to the tow "opening" effect, also see the explanation for the behaviour of this parameter given in Section 2.2.2). Additionally, the catalytic modification caused a decrease in the $R_{ct}$ parameter by approximately 147 times, while the $C_{dl}$ parameter increased by approximately 13 times at the potential of 1600 mV. Similarly, as for the HER measurements, the catalytic properties of the CF electrodes were significantly enhanced by the presence of NiFe catalyst deposits (*ca.* 13 times) rather than an increase in the electrochemically active surface area of the composite material.

**Table 6.** Electrochemical parameters for the OER, obtained at as received CF, NiFe/CF, NiCCF, and NiFe/NiCCF electrodes in contact with 0.1 M NaOH supporting solution. The results were recorded by fitting the CPE-modified Randles equivalent circuit (the superscripts attached to potential value correspond to the model from Figure 4b–d) to the experimentally obtained impedance data (reproducibility usually below 10%, $\chi^2 = 1.31 \times 10^{-6}$ to $5.37 \times 10^{-6}$).

| $E$/mV | $R_p$/Ω cm$^2$ | $C_p$/μF cm$^{-2}$ | $R_{ct}$/Ω cm$^2$ | $C_{dl}$/μF cm$^{-2}$ |
|---|---|---|---|---|
| | | **CF** | | |
| 1400 [d] | 150.7 ± 5.4 | 276.0 ± 12.4 | - | 83.29 ± 0.25 |
| 1500 [d] | 124.8 ± 7.3 | 209.0 ± 20.3 | - | 79.54 ± 0.01 |
| 1600 [b] | 197.5 ± 9.0 | 292.5 ± 25.8 | 44,694.0 ± 546.4 | 70.97 ± 0.11 |
| 1700 [b] | 193.9 ± 9.0 | 294.6 ± 25.8 | 17,117.1 ± 88.0 | 69.74 ± 0.11 |
| 1800 [b] | 158.0 ± 13.4 | 249.4 ± 42.5 | 4828.2 ± 26.4 | 71.07 ± 0.25 |
| | | **NiFe/CF** | | |
| 1400 [d] | 41.6 ± 4.9 | 1982.0 ± 479.1 | - | 276.7 ± 0.1 |
| 1500 [b] | 39.0 ± 4.3 | 2626.2 ± 427.3 | 1389.8 ± 30.3 | 959.2 ± 2.4 |
| 1600 [c] | - | - | 303.50 ± 12.9 | 917.6 ± 40.3 |
| 1700 [c] | - | - | 140.83 ± 10.5 | 859.6 ± 13.2 |
| 1800 [c] | - | - | 85.45 ± 7.9 | 740.4 ± 50.3 |
| | | **NiCCF** | | |
| 1400 [d] | 246.4 ± 12.6 | 966.81 ± 54.25 | - | 1315.5 ± 6.9 |
| 1500 [b] | 340.7 ± 9.7 | 1316.70 ± 27.82 | 1718.4 ± 37.0 | 1333.6 ± 14.1 |
| 1600 [b] | 89.2 ± 6.3 | 446.97 ± 86.27 | 386.9 ± 4.2 | 1201.5 ± 32.7 |
| 1700 [b] | 91.4 ± 13.0 | 2353.14 ± 338.99 | 202.1 ± 22.4 | 1594.5 ± 78.7 |
| 1800 [b] | 75.7 ± 12.9 | 1381.01 ± 165.61 | 110.3 ± 17.7 | 1709.7 ± 162.0 |
| | | **NiFe/NiCCF** | | |
| 1400 [d] | 199.9 ± 13.0 | 345.1 ± 64.7 | - | 1489.7 ± 135.9 |
| 1500 [b] | 392.3 ± 33.6 | 1051.3 ± 69.5 | 708.9 ± 26.1 | 2339.0 ± 100.6 |
| 1600 [b] | 81.6 ± 36.5 | 1115.8 ± 180.3 | 194.1 ± 42.8 | 1260.8 ± 10.7 |
| 1700 [b] | 15.9 ± 1.9 | 30.2 ± 5.2 | 133.8 ± 5.1 | 963.1 ± 65.6 |
| 1800 [c] | - | - | 57.9 ± 2.6 | 168.7 ± 15.4 |

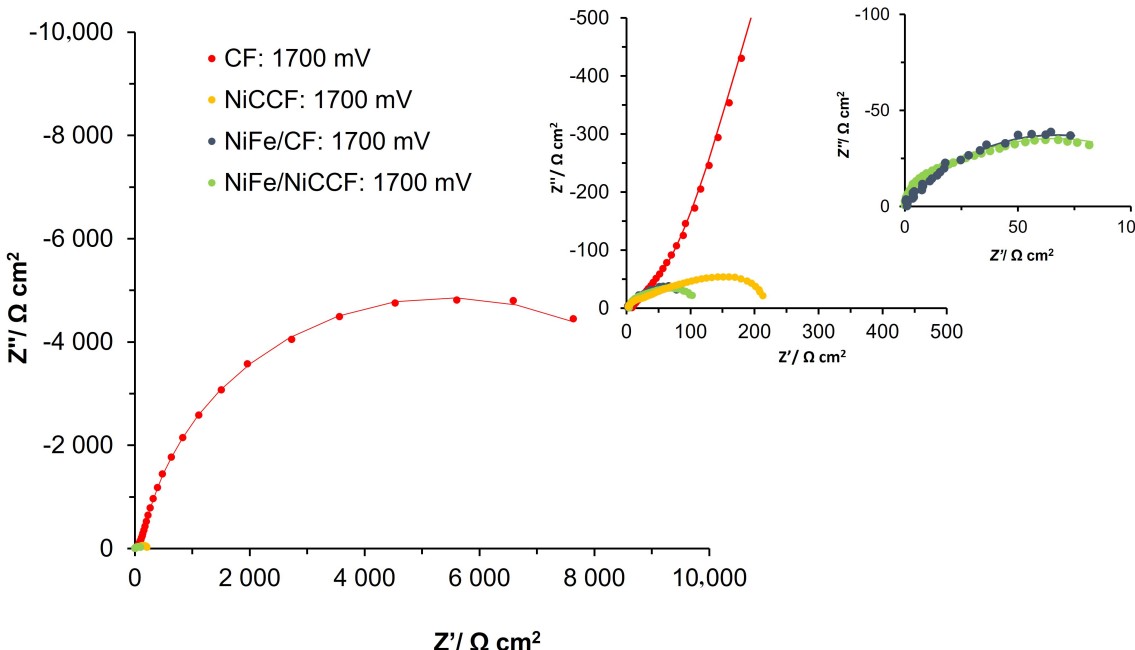

**Figure 7.** Electrochemical Nyquist impedance plots for the OER on CF, NiCCF, NiFe/CF, and NiFe/NiCCF electrode surfaces in contact with 0.1 M NaOH solution (at 293 K) for the potential of 1700 mV vs. RHE.

In the case of the catalytic modifications based on the NiCCF electrode, there were no significant differences in the $R_p$ and $C_p$ parameters between the NiCCF and the NiFe/NiCCF samples. Although the values of these parameters were somewhat fluctuating regardless of the applied potential, their values generally decreased along with increasing electrode potential. This behaviour could most likely be attributed to a more pronounced "opening" effect of the tow material by freshly-formed $O_2$ bubbles, again resulting in a loss of its somewhat porous nature.

Understandably, the $R_{ct}$ parameter for the NiCCF electrodes exhibited a significant decrease when modified with the NiFe alloy. At a potential of 1600 mV, the $R_{ct}$ value exhibited a reduction of approximately two times, indicating an improved catalytic effect of the NiFe alloy. However, it is noteworthy that the value of the $C_{dl}$ parameter remained relatively unchanged at this potential, suggesting that the surface area of the NiCCF electrode did not experience significant alteration. These findings highlight the sole catalytic effect of the NiFe modification, independent of any notable changes in the surface area, as being a primary driver behind the observed enhancement of the electrochemical performance, observed at most examined potentials. For both types of electrodes, there is a noticeable decrease in the reaction resistance as the potential increases. Specifically, for the NiCCF electrodes, the resistance decreased from 1718.4 to 110.3 $\Omega$ cm$^2$, while for the NiFe/NiCCF samples, it became reduced from 708.9 to 57.9 $\Omega$ cm$^2$ in the potential range of 1500–1800 mV. The $C_{dl}$ parameter, in this case, did not change significantly with the potential for the unmodified NiCCF electrodes, which is similar to the behaviour previously recorded for this parameter for the process of hydrogen evolution (see explanation in Section 2.2.2). However, for the NiFe/NiCCF sample, a radical decrease in the double-layer capacitance value with the rising electrode potential was observed, namely from 2339.0 to 168.7 $\mu$F cm$^{-2}$ for the potential span 1500-1800 mV. This behaviour is significantly different from that of other electrodes, probably because the effect of blocking the carbon tow's surface by the $O_2$ bubbles was considerably more prominent than an enlargement of the electrochemically accessible surface area, obtained through the physical "opening" of the tow material. Furthermore, the effect of improved OER performance is also evident in the values of the $j_0$ parameter obtained from the analysis of the Butler–Volmer equation, which

came to $2.6 \times 10^{-10}$, $9.4 \times 10^{-8}$, $9.8 \times 10^{-8}$ and $5.7 \times 10^{-7}$ A cm$^{-2}$ for the CF, NiFe/CF, NiCCF and NiFe/NiCCF samples, respectively.

### 2.2.5. Tafel-OER

Figure 8 (and corresponding Figure S7) shows the Tafel polarisation curves obtained for CF, NiFe/CF, NiCCF, and NiFe/NiCCF electrodes. The values of the anodic slope ($b_a$) and the current density at an overpotential of 0.3 V ($j_{(\eta=0.3V)}$) for the OER are provided in Table 7 (and the corresponding Table S4). The potential range for which the linear part of the plots was determined is confined to the potential range: 1500–1600 mV. Figure 9 (and corresponding Figure S8) shows that the modified samples exhibited a lower OER onset potential than that of basic materials. The current densities and $b_a$ parameters obtained for the electrodes modified with NiFe were similar to those achieved by catalysts based on noble metals, such as platinum, ruthenium, and iridium. Additionally, it can be observed in Table 7 that apart from NiFe, there are various combinations of transition metals that can exhibit similar catalytic properties. This significantly increases the number of potential catalysts that could be utilised in this application. By exploring different metal combinations, there is a possibility to discover more efficient and cost-effective catalytic materials that meet specific requirements for the examined electrochemical processes.

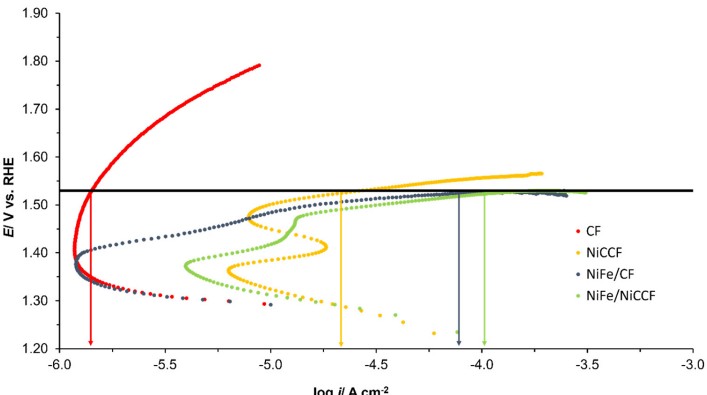

**Figure 8.** Quasi-potentiostatic cathodic polarisation curves for the oxygen evolution reaction (OER) obtained for CF, NiFe/CF, NiCCF, and NiFe/NiCCF electrodes in 0.1 M NaOH solution. The polarisation curves were recorded at a scan rate of 0.5 mVs$^{-1}$. The black line represents the overpotential of 300 mV. Arrows on the graph show the logarithm of the current density (log $j$) for each sample; the colour of the arrow matches the colour of the corresponding sample plot. The iR correction was applied to account for the solution resistance, based on the impedance measurements.

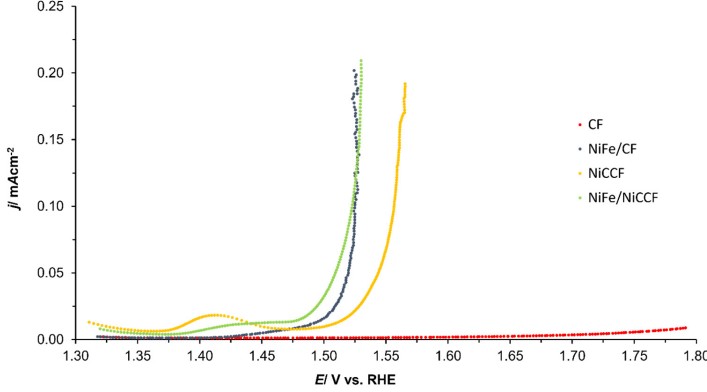

**Figure 9.** Linear Sweep Voltammetry (LSV) curves of CF, NiFe/CF, NiCCF, and NiFe/NiCCF electrodes in 0.1 M NaOH solution, carried out with a scanning rate of 0.5 mV s$^{-1}$ for OER (iR-corrected).

The current density values recorded on the examined materials at an anodic over-potential of 300 mV were similar to those achieved by bulk NiFe-LDH (Layered Double Hydroxide) materials [18]. While NiFe-modified CF and NiCCF electrodes did not exhibit as high current densities as certain other catalytic materials, such as IrO₂ or CoP, they were simple to prepare and could readily be utilised for commercial purposes [17]. However, additional research may be necessary in order to optimize their catalytic capabilities, particularly with regard to the HER activity.

Please note that the respective overpotentials for all the examined fibre-based catalysts at the current density of 10 mA cm$^{-2}$ are missing in Table 7 (also, see the results given in Figures 8 and 9). However, it has to be noted that when the catalysts' surface becomes read-justed to its electrochemically active part (see Table S4 in the supplementary information file), then the respected overpotential recorded at the current density of 10 mA cm$^{-2}$ is as follows: 560, 290, 305, and 270 mV for CF, NiFe/CF, NiCCF, and NiFe/NiCCF, correspondingly. These results are in fact fully in line (or even somewhat superior to) with those presented in Table 7 (Table S4) for other NiFe-based catalysts.

**Table 7.** OER kinetic parameters for the selected catalytic materials.

| Material | Electrolyte | $b_a$ [mV dec$^{-1}$] | $j_{(\eta=0.3V)}$ [A cm$^{-2}$] | $\eta_{(j=10mAcm^{-2})}$ [mV] | Ref. |
|---|---|---|---|---|---|
| CF | 0.1 M NaOH | 261 | $9.7 \times 10^{-6}$ | - | This work |
| NiFe/CF | 0.1 M NaOH | 40 | $9.1 \times 10^{-5}$ | - | This work |
| NiCCF | 0.1 M NaOH | 74 | $4.1 \times 10^{-5}$ | - | This work |
| NiFe/NiCCF | 0.1 M NaOH | 60 | $1.7 \times 10^{-4}$ | - | This work |
| RuO₂/GC | 0.1 M NaOH | 44 | $\sim 5.0 \times 10^{-4}$ | - | [48] |
| Co₃O₄/GC | 0.1 M KOH | 69 | $5.9 \times 10^{-6}$ | - | [49] |
| CoAl₂O₄/GC | 0.1 M KOH | 56 | $3.9 \times 10^{-7}$ | - | [49] |
| ZnCo₂O₄/GC | 0.1 M KOH | 113 | $5.6 \times 10^{-7}$ | - | [49] |
| Pt | 1.0 M KOH | 66 | $4.0 \times 10^{-4}$ | - | [12] |
| Ni/Fe | 1.0 M NaOH | 38 | $3.3 \times 10^{-5}$ | - | [16] |
| Co/Fe | 1.0 M NaOH | 46 | $1.2 \times 10^{-5}$ | - | [16] |
| IrO₂/GC | 1.0 M KOH | 76 | $3.9 \times 10^{-3}$ | - | [17] |
| CoP/C | 1.0 M KOH | 71 | $5.0 \times 10^{-3}$ | - | [17] |
| NiFe-LDH/GC | 1.0 M KOH | 35 | $\sim 9.0 \times 10^{-4}$ | 320 | [13] |
| Ni$_{0.25}$Co$_{0.75}$O$_x$ | 1.0 M KOH | 36 | $7.9 \times 10^{-5}$ | 377 | [14] |
| NiCo-LDH/GC | 1.0 M KOH | 41 | - | 335 | [18] |
| MnFe₂O₄/GC | 0.1 M KOH | 114 | - | 470 | [15] |
| NiFe₂O₄/GC | 0.1 M KOH | 98 | - | 440 | [15] |

In order to further assess the practical utilisation of our NiFe catalysts, we also con-ducted extended stability tests spanning 48 h. Figure S9 demonstrates a consistent electro-chemical performance over time, with only minor variations over the recorded cell voltage for both HER and OER processes. The voltage jump observed in the graph is attributable to the temporary halt of the experiment for carrying out EIS measurements.

## 3. Materials and Methods

### 3.1. Solutions and Chemical Reagents

All solutions were prepared using a Spring 30 s ultra-pure water purification system from Hydrolab (with a resistivity of 18.2 MΩ cm). The 0.1 M NaOH supporting solution was prepared from sodium hydroxide pellets (99.9%, POCH, Gliwice, Poland). In addition, 0.5 M H₂SO₄ was made of sulphuric acid (98% Merck, Darmstadt, Germany) for charging a palladium reversible hydrogen electrode (RHE).

### 3.2. Electrodes and Electrochemical Cell

Electrochemical experiments were carried out in a typical electrochemical cell. The cell was made of Pyrex glass and contained three electrodes: carbon fibre (CF; Hexcel 12K AS4C, 12,000 single filaments of 7 μm diameter each) or nickel-coated carbon fibre (NiCCF; Toho-Tenax 12K50, 12,000 single filaments of about 7.5 μm diameter each and ca. 45 wt.% Ni)-based working electrode (WE), Pd RHE (Pd wire, 99.99% purity, 1.0 mm diameter, Sigma-Aldrich) as reference electrode (RE) and Pt counter electrode (CE; Pt wire, 99.99% purity, 1.0 mm diameter, Sigma-Aldrich). All electrodes were placed in separate compartments. In addition, before commencing experiments, atmospheric air was removed from the cell by bubbling with argon (Ar 5.0 grade, Eurogas Bombi, Dywity, Poland). Furthermore, argon gas flow was kept above the solutions throughout the experiments.

CF/NiCCF base electrodes were 1.5 cm long, with geometrical surface areas (GSA) of 39 cm$^2$ for CF and 43 cm$^2$ for NiCCF (based on the manufacturer-provided data). In the main manuscript, we reported electrochemical parameters based on the GSA of the CF or NiCCF base electrodes. However, in order to provide a more subtle understanding of the electrochemical behaviour, we also included additional results based on the electrochemically active surface area (ECSA) in the supplementary file (SF). The ECSA, estimated from the double-layer capacitance ($C_{dl}$), as shown in Figure S1, provides a more accurate measure of the active sites available for electrochemical reactions [50,51]. Nevertheless, it was observed that the parameters based on the ECSA seemed overly optimistic, possibly due to the specific conditions or assumptions made during ECSA estimation. Therefore, to maintain a conservative and broadly comparable approach, we have chosen to present the GSA-based results in the main manuscript. By reporting both the GSA- and ECSA-based parameters, we aim to give a comprehensive view of the electrochemical performance of our electrodes, thus allowing for a more in-depth interpretation and comparison of the obtained results. In order to remove a protective epoxy resin coating, the CF samples were heat-treated (at a low oxygen atmosphere for 4 h at 623 K), while the NiCCF samples were initially de-sized in acetone. Before running the experiments, the electrodes were oxidized in 0.1 M NaOH solution at an anodic current density of 0.3 mA cm$^{-2}$ for 300 s. Catalyst electrodeposition was then performed according to the conditions and bath compositions presented in Table S5 (Supplementary Information File). Procedures for the preparation of laboratory equipment were as previously described in works by Pierożyński [26,27,29].

### 3.3. Experimental Methodology

All electrochemical measurements were performed at 293 K employing an AUTM 204 + FRA 32M Multi-Autolab potentiostat/galvanostat system. This work covers the employment of Tafel quasi-steady-state polarisation, electrochemical ac. impedance spectroscopy (EIS), and cycling voltammetry (CV) techniques. For EIS, the generator provided an output signal of a known amplitude of 5 mV, and the frequency range was typically swept between $1.0 \times 10^{-5}$ and $0.5 \times 10^{-1}$ Hz. The instruments were controlled by Nova 2.1 software for Windows (Metrohm Autolab B.V., Opacz-Kolonia, Poland). Impedance data analysis was performed with ZView 2.9 software package (Windows, Scribner Associates, Inc. Berwyn, PA, USA), where the impedance spectra were fitted with a complex, non-linear, least-squares immittance fitting program, LEVM 6, written by J.R. Macdonald [52]. For CV measurements, three sweeps were carried out over the potential span of −1.0–1.8 V vs. RHE with a scanning rate of 50 mV s$^{-1}$. Moreover, Tafel polarisation experiments (recorded at a scanning rate of 0.5 mV s$^{-1}$) for the HER and OER experiments were conducted for all examined samples. Additionally, SEM/Energy-Dispersive X-ray (EDX) surface spectroscopy characterisation of all examined CF, NiCCF, and NiFe-modified CF, and NiCCF samples was carried out by means of Merlin FE-SEM microscope (Zeiss), equipped with Bruker XFlash 5010 EDX instrumentation (with 125 eV resolution).

## 4. Conclusions

In summary, the NiFe alloy deposited at a 10% by weight via NiFe electrodeposition on CF and NiCCF serves as a promising catalyst that could potentially be used as a material for practical electrodes in alkaline water electrolysers. The incorporation of the NiFe alloy into CF entities significantly reduced (160 times at a potential of $-700$ mV vs. RHE) the HER-associated charge transfer resistance along with nearly 150 times $R_{ct}$ reduction for the corresponding OER (at a potential of 1500 mV). Similarly, for NiCCF electrodes, the reduction in the charge transfer resistance was approximately 25% and 50% for the HER and OER, respectively. These findings highlight the effectiveness of the NiFe alloy in improving the electrochemical performance of both CF and NiCCF electrodes in alkaline water electrolysis applications.

Notably, when the recorded electrochemical parameters were normalised to the electrochemically active surface area (ECSA), the values appeared significantly higher than those obtained when normalised to the geometrical surface area (GSA). This could potentially point to an exceptionally high density of active sites on the electrode surface. However, given that these ECSA-normalised values seemed overly optimistic, we chose to employ a dual approach, reporting parameters based on both the GSA and ECSA data to offer a more balanced and comprehensive understanding of the catalyst performance.

Additionally, the recorded current density levels for both hydrogen and oxygen evolution reactions indicate that the NiFe alloy deposited on carbon-based materials could serve as a cost-effective and efficient alternative to noble metals and nickel-based materials that are currently used in commercial alkaline water electrolysers. Furthermore, the development of new catalysts based on transition metal alloys will enable further increase in the water electrolysis efficiency. However, in order to design a highly catalytic, Ni-based AWE stack system for optimised production of green hydrogen, further refining of catalyst composition along with additional electrochemistry work will be required.

**Supplementary Materials:** The following supporting information can be downloaded at: https://www.mdpi.com/article/10.3390/catal13121468/s1, Figures S1 to S9; Tables S1 to S5.

**Author Contributions:** Conceptualisation, M.K.; methodology, M.K.; formal analysis, M.K.; investigation, M.K.; data curation, M.K.; writing—original draft preparation, M.K.; writing—review and editing, T.M. and B.P.; supervision, T.M. and B.P. All authors have read and agreed to the published version of the manuscript.

**Funding:** This work has been primarily financed by the internal research grant no. 30.610.001-110, provided by The University of Warmia and Mazury in Olsztyn.

**Data Availability Statement:** Data supporting reported results will be available upon request.

**Conflicts of Interest:** The authors declare no conflict of interest.

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
