# Peer review of "Alkaline Water Splitting by Ni-Fe Nanoparticles Deposited on Carbon Fibre and Nickel-Coated Carbon Fibre Substrates"

_catalysts, doi:10.3390/catal13121468_

Round 1
Reviewer 1 Report
Comments and Suggestions for Authors
The authors synthesized nickel-coated carbon fibers modified with NiFe alloy particles, and investigated the catalytic performance for water splitting in alkaline condition, both for hydrogen evolution reaction (HER) and oxygen evolution reaction (OER). The authors declared that this catalyst has much improved HER and OER activity compared to the background carbon fibers. However, the excellent performance of Ni-Fe based materials for OER and Ni-based materials for HER is well known in the research field. The performance of the catalysts in this study is actually much worse compared to the state-of-the-art Ni-Fe based catalysts. Moreover, this manuscript neither provides new insights nor develops new synthetic methods. Therefore, this Reviewer does not recommend the publication of this manuscript in its current form. The specific comments are listed as below:
1. Comprehensive and systematic electrochemical impedance data is a highlight of this research work. The Reviewer suggests the authors emphasize this part and reorganize their manuscript.
2. The design principle for nickel-coated carbon fibers modified with NiFe alloy particles is not well described in the Introduction part. What is the reason to combine Ni-coated carbon fibers and NiFe alloy together?
3. Page 5, Line 133-135. “On the other hand, peak B is related to the formation of β-NiOOH oxyhydroxide phase (Eq. 13); then peak C corresponds to its reduction (β-NiOOH/β-Ni(OH)2) [31–38].” How do the authors confirm that the peak B/C correspond to mutually transformation between β-NiOOH and β-Ni(OH)2 instead of γ-NiOOH/α-Ni(OH)2 transition?
4. For the polarization curves, including CVs and LSVs, the authors just showed the region with very small current density (below 1 mA/cm2). According to Table 7, all the catalyst can reach to at least 10 mA/cm2. The polarization curves in high current density region should also be shown to have more comprehensive understanding of the catalyst. In addition, to understand the purely electrokinetic properties, the polarization curves should be iR corrected or compensated.
5. Page 7, Line 169-170. “linear part of the plot (corresponding to mass transfer control at medium and low frequencies).” As no HER happens in the potential between -100 and -600 mV for unmodified carbon fiber electrode, why there is a mass transfer control impedance?
6. Page 9, Line 219-220. “Compared to the NiCCF electrode, the NiFe-modified CF catalyst exhibited significantly lower Rct parameter values.” This description is not correct as the Rct values of two catalysts are quite similar in a wide range of potential according to Table 3.
7. Page 10, Line 245-247. “The exchange current densities, j0 for the HER were calculated based on the Butler-Volmer equation and through the relation between the j0 and the Rct parameter for overpotential approaching 0 (see for instance equation 4 in Ref. [42]).” Please provide the details for calculating exchange current density in Experimental Section, not just simply citing previous literatures.
Comments on the Quality of English LanguageThe Quality of English Language is generally OK.
Reviewer 2 Report
Comments and Suggestions for Authors
In this work, Kuczyński et al. conducted HER and OER tests over commercially available carbon fibres and nickel-coated carbon fibres modified with nanoscale NiFe alloy particles in 0.1 M NaOH solution. They found that the NiFe-modified fibre materials were approximately 14,700% and 25% more active in the OER and HER, as compared to unmodified electrodes. These findings highlight the potential of NiFe-modified carbon fibre/nickel-coated carbon fibre materials as electrodes in industrial-scale water electrolysers. This work is recommended to publish in Catalysts after revision.
1. What is the deposition amount of NiFe in NiFe/NiCCF and NiFe/CF?
2. The NiFe particles in NiFe/NiCCF seem much larger than thoes in NiFe/CF. Do the deposition amount and the morphology of NiFe particles affect the catalytic performance?
3. Why does the NiFe/NiCCF and the NiFe/CF catalysts less active than other NiFe catalysts in Table S7?
4. Is the catalyst preparation strategy applicable to other electrochemical reactions, such as electrochemical oxidation of methanol, electrochemical reduction of CO2 or nitrate? The authors can provide a brief discussion on that (here are some references: 10.1021/acs.nanolett.2c04949, 10.1021/acscatal.2c03842, 10.1016/j.xcrp.2022.100949, 10.1039/D2EY00038E).
5. The title of this manuscript can be more concise.
6. I recommend to move some of the tables into the supporting information, as they do not provide critical values for the manuscript.
Round 2
Reviewer 1 Report
Comments and Suggestions for Authors
Regarding the revised version, I still have the following comments.
1. The Introduction Part is too long. As this is a research article instead of a review article, it is not necessary to explain the general research background and every basic concept in details. On the contrary, the design principles and the reasons for using nickel-coated carbon fibers modified with NiFe alloy particles should be well described in the Introduction (as already explained by the authors in the rebuttal letter). Therefore, it is highly recommended that the authors reorganize the Introduction Part.
2. According to the Bode’s diagram (e.g. Phys. Chem. Chem. Phys. 2013, 15, 13737), β-NiOOH/β-Ni(OH)2 and γ-NiOOH/α-Ni(OH)2 transitions can be realized by electrochemical charging/discharging. The transformation from β-NiOOH to γ-NiOOH requires overcharging (high applied potential), while transformation from α-Ni(OH)2 to β-Ni(OH)2 requires aging. As no aging was applied for the catalyst used in this study, how do authors confirm that the initial catalyst is completely in the form of β-Ni(OH)2? Actually the peaks B/C are quite broad, therefore it is probable that they can also be mixed by the features of both β-NiOOH/β-Ni(OH)2 and γ-NiOOH/α-Ni(OH)2 transitions, instead of just β-NiOOH/β-Ni(OH)2.
3. According to the Introduction, the authors aim to study the catalyst material in the condition relevant to future practical implementation. Therefore, I don’t understand why the authors just show the CV features at low current density region. The catalyst performance in high current density region should also be provided.
4. Normally electrochemical surface area (ECSA) should be bigger than the geometric area as the catalyst should be more roughened compared to atomically smooth planar surface of the corresponding material (J. Am. Chem. Soc. 2013, 135, 16977). However, in this study it looks like the calculated ECSA is smaller than the geometric area. Please check if all the related calculations are correct.
Round 3
Reviewer 1 Report
Comments and Suggestions for Authors
The authors have addressed most of my comments. However, I still have some concerns regarding the ECSA data. In the Response Letter, the authors said, “for atomically flat and smooth catalyst surfaces, their electrochemically active surface area (due to surface roughening effects) would typically exceed that of geometrically evaluated one”. Actually, the ECSA of atomically flat and smooth catalyst surfaces should be equal to their geometric area. This is the basis for the ECSA calculation (J. Am. Chem. Soc. 2013, 135, 16977−16987). On the contrary, 3-D porous catalytic structures are inherently more roughened, resulting in a higher ECSA than the geometric area. The ratio between the ECSA and the geometric area, often referred to as the roughness factor, should also exceed 1. According to the same source (J. Am. Chem. Soc. 2013, 135, 16977−16987), the ECSA should always be greater than or equal to 1 cm2, if the geometric area is 1 cm2. Therefore, it is quite confusing why the ECSA is only 0.15 cm2 in this study. Furthermore, I noted a lack of detail in Page 3 of the Supplementary Information regarding how the ECSA value of 0.15 cm2 was derived based on a capacitance of 3.25-2.79 uF.
According to the explanation in the Response Letter, the authors might have applied an alternative ECSA definition or calculation method. Nonetheless, in order to align with established literature, I recommend utilizing the standard approach outlined in the JACS paper (J. Am. Chem. Soc. 2013, 135, 16977−16987) for ECSA calculation.
